# ACCELERATING OPTIMIZATION USING NEURAL REPARAMETRIZATION

## ABSTRACT

We tackle the problem of accelerating non-linear non-convex optimization problems. We discover that reparametrizing the optimization variables as the output of a neural network can lead to significant speedup. We examine the dynamics of gradient flow of such neural reparametrization. We find that to obtain the maximum speed up, the neural network architecture needs to be a specially designed graph convolutional network (GCN). The aggregation function of the GCN is constructed from the gradients of the loss function and reduces to the Hessian in early stages of the optimization. We show the utility of our method on two optimization problems: network synchronization and persistent homology optimization, and find an impressive speedup, with our method being $4 \sim 80\times$ faster.

## 1 INTRODUCTION

Gradient-based optimization lies at the heart of machine learning. For non-linear, non-convex large-scale optimization problems, gradient-based optimization can suffer from slow convergence and significant computational bottleneck. In linear systems, *preconditioning* Axelsson (1996); Saad & Van Der Vorst (2000) can accelerating convergence by multiplying a symmetric positive-definite preconditioner matrix to the original problem. In effect, preconditioning re-scales the loss landscape to have more circle-like level sets, hence changing the condition number of the system. Adaptive gradient optimizers such as AdaGrad Duchi et al. (2011) and Adam Kingma & Ba (2014) uses the same idea but updates the preconditioner during iterative optimization with adaptive step size, which leads to great improvement in convergence speed.

Our goal is to improve the convergence of non-convex optimization further. First, we examine the dynamics of the gradient flow and identify the condition number that dictates the speed of convergence. We discover that changing the condition number naturally requires the reparametrization to be a type of graph neural networks (GNN). Based on the analysis, we introduce a novel neural reparametrization scheme, which generalizes preconditioning to non-linear systems. Note that our method is complementary to existing gradient-based optimization algorithms and can be used in conjunction with *any* adaptive gradient optimizer (e.g. Adam). We test our hypothesis on two highly nonlinear optimization problems and obtain impressive speedups.

A few earlier work Sosnovik & Oseledets (2019); Hoyer et al. (2019) have demonstrated that convolutional neural networks can serve as priors to improve the parameterization of a class of structural optimization problems. But the theoretical foundation behind these techniques are not well understood. Concurrently, the *implicit acceleration* of over-parametrization in linear neural networks have been analyzed in Arora et al. (2018); Tarmoun et al. (2021). Specifically, Arora et al. (2018) shows that reparametrizing the linear weights with deep linear networks impose a pre-conditioning scheme on gradient descent. Tarmoun et al. (2021) further exploits the conservation laws in gradient flow and derive convergence rate for two-layer linear networks.

In contrast, our method applies to highly nonlinear problems, guided by theoretical insights. Our contributions can be summarized as follows

1. We analyze the dynamics of the gradient flow and derive how a neural reparametrization (NR) can lead to a speedup.
2. We prove that maximum speedup is achieved if the NR is a special GCN which uses gradient of the loss in its aggregation function.

3. For early steps, we show that the NR should use the Hessian of the loss, leading to an efficient hybrid implementation.

4. We showcase NP on two optimization tasks: network synchronization and persistent homology and observe consistent speedup.

Our method also demonstrates a novel usage and unique perspective for GNN, specifically GCN (Kipf & Welling, 2016). The majority of literature treat GNNs as a representation learning tool for graph-structure data (see surveys and the references in Bronstein et al. (2017); Zhang et al. (2018); Wu et al. (2019); Goyal & Ferrara (2018)). However, our analysis shows that GNN can be used to modify optimization dynamics in problems *without* explicit graph data.

## 2 METHODOLOGY

We consider the problem of minimizing a smooth, lower-bounded loss function $\mathcal{L}(\boldsymbol{w}) \in \mathbb{R}$ on abounded variable $\boldsymbol{w} \in \mathbb{R}^n$:

$$\mathrm{argmin}_{\boldsymbol{w}} \mathcal{L}(\boldsymbol{w})$$

When the optimization variables are matrices or tensors, $\boldsymbol{w}$ represents a flattened vector containing all variables. $\boldsymbol{w}$ can also represent all trainable parameters in a deep neural network. For supervised learning with a dataset $\boldsymbol{Z} = \{(\mathbf{x}_i, \mathbf{y}_i)_{i=1}^N\}$, the loss function $\mathcal{L}(\boldsymbol{w})$ becomes $\mathcal{L}(\boldsymbol{w}; \boldsymbol{Z})$. The dataset and the model class (e.g. architecture of a neural network) together define a loss landscape as a function of the optimization variables $\boldsymbol{w}$.

Gradient-based optimization updates the variables $\boldsymbol{w}$ by taking repeated steps in the direction of the steepest descent $\boldsymbol{w}^{t+1} = \boldsymbol{w}^t - \varepsilon \frac{\partial \mathcal{L}}{\partial \boldsymbol{w}^t}$ for a learning rate $\varepsilon$. With very small time steps $\delta t$, we examine the dynamics of $\boldsymbol{w}(t) \in \mathbb{R}^n$ during the optimization process, treating $t$ as a continuous time. Using steepest descent for the optimization, $d\boldsymbol{w}/dt$ follows the gradient flow:

$$\frac{d\boldsymbol{w}_i}{dt} = -\varepsilon_{ii} \frac{\partial \mathcal{L}}{\partial \boldsymbol{w}_i}, \tag{1}$$

where denote the learning rate $\varepsilon$ as a diagonal matrix of learning rates, because modern optimizers such as RMSProp (Tieleman et al., 2012) or Adam (Kingma & Ba, 2014) choose $\varepsilon$ adaptively for each $\boldsymbol{w}_i$. Specifically, in Adam and RMSProp, which we use in experiments, we have

$$\varepsilon_{ij}(\nabla \mathcal{L}, t) \approx \frac{\eta}{\sqrt{\mathbb{E}_t[(\nabla_i \mathcal{L})^2] + \xi}} \delta_{ij} \tag{2}$$

where $\delta_{ij}$ is the Kronecker delta, and $\eta$ and $\xi$ are small numbers. The expected value $\mathbb{E}$ is calculated using stochastic samples for $\nabla_i \mathcal{L}$ over multiple iteration steps, with more weights given to more recent steps. Equation 1 is also an ordinary differential equation (ODE). The local minima of $\mathcal{L}(\boldsymbol{w})$ correspond to the stationary solutions of the ODE, i.e. $d\boldsymbol{w}/dt = 0$.

**Convergence rate**  We are interested in the expected convergence rate $\mathbb{E}[d\mathcal{L}/dt]$ of the loss function. This expectation can be either approximated in a fashion similar to adaptive gradient optimizers (i.e. exponential time average) or be over random initializations under a distribution $P(\boldsymbol{w}(0))$. We will assume the expectation value for $\varepsilon$ and $\mathbb{E}[d\mathcal{L}/dt]$ are evaluated with the same method, meaning we assume $\mathbb{E}[\varepsilon] = \varepsilon$. The convergence rate is then given by

$$\mathbb{E}\left[\frac{d\mathcal{L}(\boldsymbol{w})}{dt}\right] = \mathbb{E}\left[\sum_i \frac{\partial \mathcal{L}}{\partial \boldsymbol{w}_i} \frac{d\boldsymbol{w}_i}{dt}\right] = -\mathrm{Tr}\left[\varepsilon \overline{M}\right], \qquad \overline{M}_{ij} \equiv \mathbb{E}\left[\frac{\partial \mathcal{L}}{\partial \boldsymbol{w}_i} \frac{\partial \mathcal{L}}{\partial \boldsymbol{w}_j}\right] \tag{3}$$

where we used our assumption to get $\mathbb{E}[\varepsilon M] = \varepsilon \mathbb{E}[M]$, with $M_{ij} = \nabla_i \mathcal{L} \nabla_j \mathcal{L}$ being the "square gradients" matrix and $\overline{M} = \mathbb{E}[M]$. With this definition, equation 2 becomes $\varepsilon = \eta / \sqrt{\mathrm{diag}[\overline{M}] + \xi}$. Note that $\overline{M}$ is time-dependent and can change during optimization.

**Estimating $\overline{M}$**  If we know the distribution $P(\boldsymbol{w}(t))$ for some $t$, we can approximate $\overline{M}$. One way to do this is similar to Adam as a weighted average over the past few steps. Also, if we know the distribution of the initialization $P(\boldsymbol{w}(0))$, we can evaluate $\overline{M}$ for early stages, which is what we

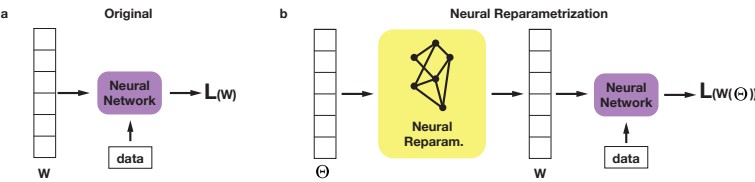

Figure 1: Original (a) vs Neural Reparametrization (b) of the optimization problem.

will use in the current paper. When using Xavier initialization (Glorot & Bengio, 2010), we have $\boldsymbol{w}(0) \sim \mathcal{N}(0, n^{-1/2})$. In this case we can show (see appendix A.1)

$$\overline{M}_{ij}(t \to 0) = \sum_{k,l} \mathbb{E}[\boldsymbol{w}_k \boldsymbol{w}_l] \frac{\partial^2 \mathcal{L}}{\partial \boldsymbol{w}_k \partial \boldsymbol{w}_i} \frac{\partial^2 \mathcal{L}}{\partial \boldsymbol{w}_l \partial \boldsymbol{w}_j} + O(n^{-2}) = \frac{1}{n} \left[ \mathcal{H}^2 \right]_{ij} \Big|_{\boldsymbol{w} \to 0} + O(n^{-2}), \quad (4)$$

where $\mathcal{H}_{ij}(\boldsymbol{w}) \equiv \partial^2 \mathcal{L}(\boldsymbol{w})/\partial \boldsymbol{w}_i \partial \boldsymbol{w}_j$ is the Hessian of the loss at $t = 0$.

The eigenvectors of $\overline{M}$ with zero eigenvalue are modes that do not evolve during optimization. As gradients are zero in these directions, GD can never find and explore those directions. Therefore, we may exclude them from the beginning and claim that $\overline{M}$ is full rank.

The rate $d\boldsymbol{w}_i/dt$ can be different along different directions of the parameter space, with fast and slow modes. Next, we quantify this difference in rates and suggest ways for acceleration.

## 2.1 FAST AND SLOW DYNAMICS OF GRADIENT FLOW

$\overline{M}$ is positive semi-definite because for any vector $\boldsymbol{v} \in \mathbb{R}^n$ we have $\boldsymbol{v}^T \overline{M} \boldsymbol{v} = \mathbb{E}\left[ \left( \boldsymbol{v}^T \nabla \mathcal{L} \right)^2 \right] \geq 0$. Thus, $\overline{M}$ is Hermitian with a spectral expansion $\overline{M} = \sum_i m_i \psi_i \psi_i^T$. Using equation 1, we can show that the dynamics of $\boldsymbol{w}$ along different $\psi_i$ are orthogonal to each other

$$\mathbb{E}\left[ \left( \psi_i^T \frac{d\boldsymbol{w}}{dt} \right) \left( \psi_j^T \frac{d\boldsymbol{w}}{dt} \right) \right] = \varepsilon^2 \psi_i^T \overline{M} \psi_j = \varepsilon_i \varepsilon_j m_i \delta_{ij}. \quad (5)$$

If $\varepsilon$ is constant and not adaptive, $\boldsymbol{w}$ evolves faster along modes $\psi_i$ with larger eigenvalues $m_i$, since if $m_i > m_j$ we have

$$\varepsilon^2 m_i = \mathbb{E}\left[ \left( \psi_i^T \frac{d\boldsymbol{w}}{dt} \right)^2 \right] > \mathbb{E}\left[ \left( \psi_j^T \frac{d\boldsymbol{w}}{dt} \right)^2 \right] = \varepsilon^2 m_j \quad (6)$$

This is precisely what optimizers such as Adam try to address. Ideally, having $\varepsilon_i = \eta/\sqrt{m_i}$ would make up for the rate difference, resulting in a uniform GD where no mode $\psi_i$ is evolving more slowly than others. However, calculating $m_i$ is $O(n^3)$ and can be quite expensive during optimization, hence the approximate version equation 2 is used. We show here that in many cases it is worthwhile to do a full correction to GD using $\overline{M}$, at least in early stages of the optimization.

The eigenvalues of $\overline{M}$ determine the rate of evolution of the overlaps along each of its eigenvectors. In particular, if some eigenvalue $m_{slow} \ll \text{mean}_i[m_i]$ is much smaller than the mean, the evolution of $\boldsymbol{w}$ along $\psi_{slow}$ will be much slower than other directions. Therefore, we will refer to all $\psi_{slow}$ as the *slow modes* of the optimization problem. Conversely, the $\psi_{fast}$ where $m_{fast} \gg \text{mean}_i[m_i]$ will be referred to as the *fast modes*.

When running GD, the maximum change in $\boldsymbol{w}$ is bounded to ensure numerical stability. Because of the orthogonality in equation 5, we can enforce a numerical bound $\eta$ via

$$\max_i \mathbb{E}\left[ \left( \psi_i^T \frac{d\boldsymbol{w}}{dt} \right)^2 \right] \leq \eta^2 \Rightarrow \varepsilon_{max} \leq \frac{\eta}{\sqrt{m_{max}}}, \qquad \mathbb{E}\left[ \left( \psi_i^T \frac{d\boldsymbol{w}}{dt} \right)^2 \right] \leq \eta^2 \frac{m_i}{m_{max}}. \quad (7)$$

where $m_{max} = \max_i m_i$ is the largest eigenvalue of $M$. Therefore, the learning rate $\varepsilon$ is bounded by the largest eigenvalues (fastest modes). Thus, in order to speed up *convergence*, we need to focus on the slow modes, which can be achieve with a reparametrization of $\boldsymbol{w}$.

## 2.2 NEURAL PARAMETRIZATION OF OPTIMIZATION PROBLEMS

Using neural networks to reparametrize optimization problems have been suggested in (Hoyer et al., 2019; Sosnovik & Oseledets, 2019). The idea is that the optimization variable $\boldsymbol{w}$ can be defined as the output of a deep neural network $\boldsymbol{w}(\theta) \equiv f(\theta)$ with trainable parameters $\theta$. Then, rather than optimizing over the original variables $\boldsymbol{w}$ directly, we can instead optimize over the neural network parameters $\theta$. We will take this idea further and analyze the type of neural network architectures that can accelerate the optimization.

**Modified convergence rate**  The neural reparametrization of the variables $\boldsymbol{w}(\theta)$ leads the following updating rule for $\theta$:

$$\frac{d\theta_a}{dt} = -\hat{\varepsilon}_a \frac{\partial \mathcal{L}}{\partial \theta_a} = -\hat{\varepsilon}_a \sum_i \frac{\partial \mathcal{L}}{\partial \boldsymbol{w}_i} \frac{\partial \boldsymbol{w}_i}{\partial \theta_a} \tag{8}$$

The convergence rate is then given by

$$\frac{d\mathcal{L}}{dt} = \sum_{i,a} \frac{\partial \mathcal{L}}{\partial \boldsymbol{w}_i} \frac{\partial \boldsymbol{w}_i}{\partial \theta_a} \frac{d\theta_a}{dt} = -\sum_{i,j} \frac{\partial \mathcal{L}}{\partial \boldsymbol{w}_i} \frac{\partial \mathcal{L}}{\partial \boldsymbol{w}_j} \sum_a \varepsilon_a \frac{\partial \boldsymbol{w}_i}{\partial \theta_a} \frac{\partial \boldsymbol{w}_j}{\partial \theta_a}$$

$$= -\sum_{i,j} M_{ij} K_{ij} = -\operatorname{Tr}\left[MK^T\right], \qquad K_{ij} \equiv \sum_a \hat{\varepsilon}_a \frac{\partial \boldsymbol{w}_i}{\partial \theta_a} \frac{\partial \boldsymbol{w}_j}{\partial \theta_a}. \tag{9}$$

Here $K$ is the neural tangent kernel (NTK) (Jacot et al., 2018) of $\boldsymbol{w}(\theta)$, and $M_{ij}$ is the squared gradients. Note, we have absorbed $\hat{\varepsilon}_a$ into $K$ for convenience of notation.

**Expected convergence rates**  The expected convergence rate $\mathbb{E}[d\mathcal{L}/dt]$ equation 3 depended on $\overline{M} = \mathbb{E}[M]$, the reparametrized one from equation 9 depends on $\mathbb{E}[MK]$. In order to quantify any improvements in $\mathbb{E}[d\mathcal{L}/dt]$ from equation 9, we need to be able to express $\mathbb{E}[MK]$ in terms of $\mathbb{E}[M]$, which is not possible for a general $K$.

If we assume $M$ and $K$ to be independent from each other, i.e. $\mathbb{E}[MK]$ becomes $\mathbb{E}[M]\mathbb{E}[K]$. Since $M$ is a function of $\boldsymbol{w}$, $K$ must *not* be a function of $\boldsymbol{w}$. One way is to have $\boldsymbol{w} = \sigma(A\theta + b)$ or any other similar neural network. We will discuss explicit choices for $\boldsymbol{w}$ below in sec. 2.3. In summary, an important constraint that we will put on $\boldsymbol{w}(\theta)$ is that it yields a $K$ independent of $\boldsymbol{w}$

$$\text{choose } \boldsymbol{w}(\theta) \text{ s.t.:} \qquad \mathbb{E}[MK] = \mathbb{E}[M]\mathbb{E}[K] \tag{10}$$

Now we can compute the expected rate of change of $\boldsymbol{w}(\theta)$ along the previous fast and slow mode eigenvectors $\psi_i$ of $\overline{M}$, similar to equation 6. Using equation 9 and equation 8 we have

$$\psi_i^T \frac{d\boldsymbol{w}}{dt} = \psi_i^T \sum_a \frac{\partial \boldsymbol{w}}{\partial \theta_a} \frac{d\theta_a}{dt} = -\psi_i^T \sum_a \hat{\varepsilon}_a \frac{\partial \boldsymbol{w}}{\partial \theta_a} \sum_j \frac{\partial \boldsymbol{w}_j}{\partial \theta_a} \frac{\partial \mathcal{L}}{\partial \boldsymbol{w}_j} = -\psi_i^T K \frac{\partial \mathcal{L}}{\partial \boldsymbol{w}} \quad (\in \mathbb{R}) \tag{11}$$

where $\partial \mathcal{L}/\partial \boldsymbol{w} = \nabla_{\boldsymbol{w}} \mathcal{L}$ is the gradient vector. Multiplying equation 11 by its transpose, denoting $\overline{K} \equiv \mathbb{E}[K]$, and using equation 10 the expected value becomes

$$\mathbb{E}\left[\left(\psi_i^T \frac{d\boldsymbol{w}}{dt}\right)^2\right] = \psi_i^T \overline{K}\,\overline{M}\,\overline{K}^T \psi_i = \sum_j m_j \left(\psi_i^T \overline{K} \psi_j\right)^2. \tag{12}$$

which shows how the neural reparametrization may speed up the convergence.

First, note that NR changes the rate of evolution of modes $\psi_i$ in equation 7. We can quantify this change when $[\overline{K}, \overline{M}] = 0$. In this case, $\overline{K}$ and $\overline{M}$ can be diagonalized simultaneously with shared eigenvectors $\psi_i$. Plugging the spectral expansion $K = \sum_i k_i \psi_i \psi_i^T$ into equation 12 and using $\varepsilon < \eta/\sqrt{m_{max}}$ (equation 7) we have

$$\text{if: } [\overline{K}, \overline{M}] = 0 \qquad \Rightarrow \qquad \mathbb{E}\left[\left\|\psi_i^T \frac{d\boldsymbol{w}}{dt}\right\|^2\right] = k_i^2 m_i < \eta^2 \frac{m_i k_i^2}{m_{max}}. \tag{13}$$

Thus, one way to speed up the convergence, is to have $[\overline{K}, \overline{M}] = 0$ and have $\overline{K}$ increase the slowest rates $\|\psi_{slow} d\boldsymbol{w}/dt\|$, while keeping the fastest rate mostly unchanged. As a result, we have:

**Theorem 1.** *With a neural reparametrization yielding $\left[\overline{K}, \overline{M}\right] = 0$, to make all modes $\psi_i$ reach the ideal maximum rate $\max_i \|\psi_i^T d\boldsymbol{w}/dt\| < \eta$, we need to have*

$$\overline{K} \approx \left(\frac{\overline{M}}{m_{max}}\right)^{-1/2} \tag{14}$$

*Proof.* This is because plugging $k_i^2 \approx m_{max}/(m_i + \xi)$ ($\xi \ll 1$) into equation 13 makes the r.h.s. equal to $\eta^2$. $\qquad\square$

Next, we discuss how we can achieve this in practice. Specifically, we derive what architecture for the neural network $\boldsymbol{w}(\theta) = f(\theta)$ will result in a $\overline{K}$ satisfying the condition in equation 14.

## 2.3 Practical Implementation

To find a neural network model that satisfies equation 14, we need to consider the fact that $\overline{M}(\boldsymbol{w}(t))$ changes during optimization. To minimize the overhead from constructing $\boldsymbol{w}(\theta)$ (and hence $\overline{K}$) as a function of $\overline{M}$, the architecture has to be as simple as possible and involve steps which do not increase the time complexity of each iteration significantly. We will now discuss when such neural reparametrization is feasible and how to implement equation 14 in practice.

**Hybrid optimization** To use equation 14 at points other than the initial steps, we need to dynamically estimate and update $\overline{M}$. Though estimating $\overline{M}(t)$ efficiently with ideas from Adam is possible, in the current work we focus on the speeding up at early stages using equation 4. Thus, our main proposal is to use a two stage hybrid optimization: 1) use the Hessian to reparametrize $\boldsymbol{w}(\theta)$ in early stages for some number of iterations and then 2) switch to the original optimization over $\boldsymbol{w}$.

**Simple implementation** Because $\overline{M}$ may depend on $\boldsymbol{w}(\theta)$, as in equation 10, we need a $K$ that is independent of $\theta$ to ensure $\mathbb{E}[MK] = \mathbb{E}[M]\mathbb{E}[K]$. The simplest architecture is a linear model $\boldsymbol{w}_i = \sum_a \kappa_{ia}\theta_a$, which yields $\overline{K} = \kappa\kappa^T \approx (\overline{M}/m_{max})^{-1/2}$. We need $\overline{K}$ to be a full rank matrix, meaning that $\kappa$ needs to be at least $n \times n$. Since we need the least computationally expensive $\overline{K}$, we choose $\theta \in \mathbb{R}^n$, and let $\kappa$ be a symmetric $n \times n$ matrix. This way, $\kappa = \sqrt{\overline{K}} = (\overline{M}/m_{max})^{-1/4}$. For the initial stages, where $\overline{M} \approx \mathcal{H}^2/n$ we have $\kappa = (\mathcal{H}/h_{max})^{-1/2}$, with $\mathcal{H}$ being the Hessian.

**Per-step time complexity** Since $\boldsymbol{w} \in \mathbb{R}^n$, the complexity of computing $\mathcal{L}(\boldsymbol{w})$ is $O(np)$ where $p$ depends on the sparsity structure of $\mathcal{L}$. $\overline{M}$ is an $n \times n$ matrix and computing $\overline{M}^{-1/2}$ is $O(n^3)$, which is very expensive when $n \gg 1$. The eigenvalue $m_{max}$ can be estimated iteratively in $O(qn^2)$. Hence, computing $\overline{K}$ by exactly following equation 14 is not feasible, unless $\overline{M}$ is fixed and can be calculated offline. Even then, evaluating $\overline{M}^{-1/2}$ only makes sense if the expected GD steps for convergence is more than $O(n)$, otherwise we don't gain any speedup. We may instead try to approximate $M^{-1/2}$ and $m_{max}$ iteratively with complexity $O(qn^2)$ where $q \ll n$ is the number of iterations. Hence we note that using equation 14 to speedup the optimization may generally be useful if the the per step complexity of GD on $\mathcal{L}$ has a time complexity of least $O(qn^2)$.

**Implementation constraints** In summary, we have the following constraints on the structure of the neural network $\boldsymbol{w}(\theta)$

1. Computing $\boldsymbol{w}(\theta) \in \mathbb{R}^n$ must be at most $O(qn^2)$ with $q \ll n$

2. To have $\overline{K}$ independent of $\overline{M}$, $\partial\boldsymbol{w}/\partial\theta$ must be independent of $\boldsymbol{w}$ and hence, only linear maps $\boldsymbol{w}(\theta) = \kappa\theta$ are allowed, with $\kappa$ independent of $\boldsymbol{w}$.

3. $\theta \in \mathbb{R}^m$ with $m \geq n$. We choose $m = n$, which yields $\kappa = \sqrt{\overline{K}} = (\overline{M}/m_{max})^{-1/4}$

4. Instead of $\kappa = (\overline{M}/m_{max})^{-1/4}$, which is $O(n^3)$, we should use $O(qn^2)$ approximations.

**Relation to GCN** Since both $\theta \in \mathbb{R}^n$ and $\boldsymbol{w} \in \mathbb{R}^n$, our linear $\boldsymbol{w} = \kappa\theta$ architecture is essentially a Graph Convolutional Network (GCN) (Kipf & Welling, 2016) with the aggregation rule $\kappa = (\overline{M}/m_{max})^{-1/4}$, or a weighted adjacency matrix. In fact, all of our derivations remain unchanged if $\boldsymbol{w} \in \mathbb{R}^{n \times d}$ and $\theta \in \mathbb{R}^{n \times m}$ (i.e. if there are $d$ and $m$ features per node in $\boldsymbol{w}$ and $\theta$). Thus, our claim is that we can speed up the optimization process by reparametrizing $\boldsymbol{w}$ using a GCN with linear activation, as $\boldsymbol{w} = \kappa\theta$, where $\theta$ are trainable and $\kappa$ is the aggregation function derived from the loss gradients. However, evaluating this $\kappa$ during GD can be quite expensive ($O(n^3)$) for large $n$ (GD steps $\sim O(n^2)$). Ideally we want an approximation for $\kappa$ which is $O(qn^2)$ with $q \sim O(1)$.

**Computationally efficient implementation** We will focus on speeding up the early stages, in which we use the Hessian $\mathcal{H}$ in $\overline{M} \approx \mathcal{H}^2/n$. Define $\boldsymbol{H} \equiv (1-\xi)\mathcal{H}/h_{max}$, where $h_{max} = \sqrt{m_{max}}$ and $\xi \ll 1$. To get an $O(qn^2)$ approximation for $\kappa = (\overline{M}/m_{max})^{-1/4} = \boldsymbol{H}^{-1/2}$ we can take the first $q$ terms in the binomial expansion (see Appendix A.2). A rough estimate would be

$$\boldsymbol{H}^{-1/2} \approx I - \frac{1}{2}(I - \boldsymbol{H}) - \frac{3}{4}(I - \boldsymbol{H})^2. \tag{15}$$

Since $\boldsymbol{H}$ is positive semi-definite and its largest eigenvalue is $1 - \xi < 1$, the sum in equation 15 can be truncated after $q \sim O(1)$ terms. The first $q$ terms of equation 25 can be implemented as a $q$ layer GCN with aggregation function $f(\overline{M}) = I - M_0$ and residual connections. We choose $q$ to be as small as 1 or 2, as larger $q$ may actually slow down the optimization. Note that computing $f(M_0)^q\theta$ is $O(qn^2)$ because we do not need to first compute $f(\overline{M})^q$ (which is $O(qn^3)$) and instead use $\boldsymbol{v}_{i+1}f(\overline{M})\boldsymbol{v}_i$ ($O(n^2)$) $q$ times with $\boldsymbol{v}_1 = \theta$.

Lastly, to evaluate $M_0$ we need to estimate the leading eigenvalue $m_{max}$. Given any vector $\boldsymbol{v} \in \mathbb{R}^n$ and using spectral expansion, we have

$$\mathcal{H}^q\boldsymbol{v} = \sum_i m_i^{q/2}(\psi_i^T\boldsymbol{v})\psi_i \qquad \Rightarrow \qquad h_{max} \approx \frac{\boldsymbol{1}^T\mathcal{H}^2\boldsymbol{1}}{\boldsymbol{1}^T\mathcal{H}\boldsymbol{1}} = \frac{\sum_{ij}\mathcal{H}_{ij}^2}{\sum_{ij}\mathcal{H}_{ij}} \tag{16}$$

where, since $\mathcal{H}$ is positive semi-definite, for $q > 1$ the leading eigenvector $\psi_{max}$ quickly dominates the sum and we have $\mathcal{H}^q\boldsymbol{v} \approx h_{max}^q(\psi_{max}\boldsymbol{v})\psi_{max}$. Here we chose $q = 2$ to get a crude approximation for $h_{max}$. The generalized Perron-Frobenius theorem (Berman & Plemmons, 1994), state the components of the leading eigenvector $\psi_{max}$ should be mostly positive. Therefore, we chose $\boldsymbol{v} = \boldsymbol{1}/\sqrt{n}$, where $\boldsymbol{1} = (1, \ldots, 1)$, which should be close to the actual $\psi_{max}$.

If we think of $M$ as the weighted adjacency matrix of a graph with $n$ vertices, the vector component $D_i = [\mathcal{H}\boldsymbol{1}]_i = \sum_j \mathcal{H}_{ij}$ is the weighted degree of node $i$. Hence equation 16 states $\mathcal{H}/m_{max} \approx \left(\sum_i D_i/\|D\|^2\right)\mathcal{H}$. This is similar to the graph diffusion operator $D^{-1}\mathcal{H}$ and $D^{-1/2}\mathcal{H}D^{-1/2}$ which are used as the aggregation functions in GCN (here $D_{ij} = D_i\delta_{ij}$ is the degree matrix).

When using modern optimizers, the adaptove learning rates is a term within $\overline{K}$. This will interfere with the optimal choice for $\kappa$ and hence our GCN aggregation function. Therefore, in practice, we do not fix the coefficients for $\mathcal{H}$ and $\mathcal{H}^2$ in our GCN. Instead, we use single or two layer GCN with aggregation function $f(\mathcal{H}) = D^{-1/2}\mathcal{H}D^{-1/2}$.

## 3 EXPERIMENTS

We showcase the acceleration effect of neural reparametrization on two non-convex optimization problems: network of Kuramoto oscillators which are widely-used for modeling synchronization phenomena; and persistent homology, a mathematical tool that computes the topology features of data that persist across scales . We use Adam Kingma & Ba (2014) optimizer and compare different reparameterization models. We implemented all models with Pytorch.

### 3.1 NETWORK OF KURAMOTO OSCILLATORS

*Kuramoto model* Kuramoto (1975; 1984) are widely used for synchronization problems, which have profound impact on engineering, physics and social systems (Pikovsky et al., 2003). As shown in Fig. 3(right), Kuramoto model describes the behavior of a large set of coupled oscillators. Each

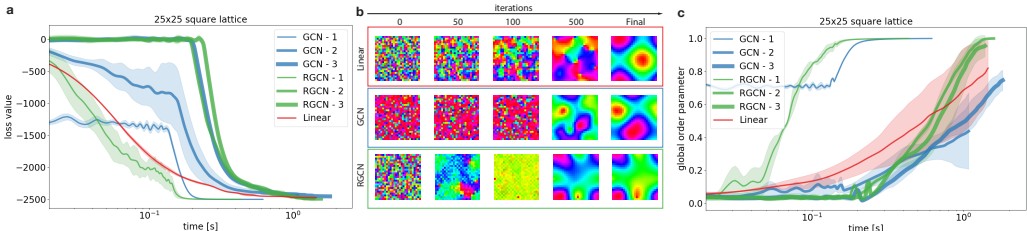

Figure 2: Optimizing Kuramoto model on a $25 \times 25$ square lattice (a) Loss over run time in seconds for different methods. (b) Evolution of the phase variables over iterations. (c) Level of synchronization, measured by global order parameter over time. Neural reparameterization with GCN achieves the highest speedup.

oscillator is defined by an angle $\theta_i = \omega_i t + \phi_i$, where $\omega_i$ is the frequency and $\phi_i$ is the phase. We consider the case where $\omega_i = 0$. The coupling strength is represented by a graph adjacency matrix with elements $A_{ij} \in \mathbb{R}$. Defining $\Delta_{ij} \equiv \phi_i - \phi_j$, the dynamics of the phases $\phi_i(t)$ in the Kuramoto model follows

$$\frac{d\phi_i}{dt} = -\varepsilon \sum_{j=1}^{n} A_{ji} \sin \Delta_{ij}, \qquad \mathcal{L}(\phi) = \sum_{i,j=1}^{n} A_{ji} \cos \Delta_{ij}. \qquad (17)$$

*Hopf-Kuramoto* (HK) model Lauter et al. (2015) is a more general version of the Kuramoto model. It includes second neighbor interactions. HK has a rich phase space with qualitatively different solutions. The phase space includes regions where simulations becomes slow and difficult. We use the HK model to showcase our method in more complex and nonlinear scenarios. The HK model's dynamics follows

$$\frac{d\phi_i}{dt} = c \sum_{j} A_{ji} \left[ \cos \Delta_{ij} - s_1 \sin \Delta_{ij} \right]$$
$$+ s_2 \sum_{k,j} \left( A_{ij} A_{jk} \left[ \sin \left( \Delta_{ji} + \Delta_{jk} \right) - \sin \left( \Delta_{ji} - \Delta_{jk} \right) \right] + A_{ij} A_{ik} \sin \left( \Delta_{ji} + \Delta_{ki} \right) \right). \quad (18)$$

with the following loss function

$$\mathcal{L} = \frac{c}{\varepsilon} \sum_{i,j} A_{ji} \left[ \sin \Delta_{ij} + s_1 \cos \Delta_{ij} \right] + \frac{s_2}{2\varepsilon} \sum_{i,k,j} A_{ij} A_{jk} \left[ \cos \left( \Delta_{ji} + \Delta_{jk} \right) + \cos \left( \Delta_{ji} - \Delta_{jk} \right) \right]$$
$$(19)$$

Our goal is to minimize the phase drift $d\phi_i/dt$ to synchronize the oscillators. Existing numerical methods directly optimize the loss $\mathcal{L}(\phi)$ with gradient-based algorithms, which we refer to as *linear*. We apply our method to reparametrize the phase variables $\phi$ and speed up convergence towards synchronization.

**Implementation.** For early stages, we use a GCN with the aggregation function derived from the Hessian which for the Kuramoto model simply becomes $\mathcal{H}_{ij}(0) = \partial^2 \mathcal{L}/\partial \phi_i \partial \phi_j |_{\phi \to 0} = A_{ij} - \sum_k A_{ik} \delta_{ij} = -L_{ij}$, where $L = D - A$ is the graph Laplacian of $A$. We found that NR in the early stages of the optimization gives more speed up. We implemented the hybrid optimization described earlier, where we reparemtrize the problem in the first 100 iterations and then switch to the original *linear* optimization for the rest of the optimization.

We experimented with three Kuramoto oscillator systems with different coupling structures: square lattice, circle graph, and tree graph. For each system, the phases are randomly initialized between $0$ and $2\pi$ from uniform distribution. We let the different models run until the loss converges (10 patience steps for early stopping, $10^{-15}$ loss fluctuation limit).

**Results of Kuramoto Model.** Figure 2 shows the results of Kuramoto model on a square lattice. Additional results on circle graph, and tree graph can be found in Appendix B. Figure 2 (a) shows that NR with one-layer GCN (GCN-1) and GCN with residual connection (RGCN-1) achieves significant speedup. In particular, we found $3.6 \pm .5$ speed improvement for the lattice, $6.1 \pm .1$ for the

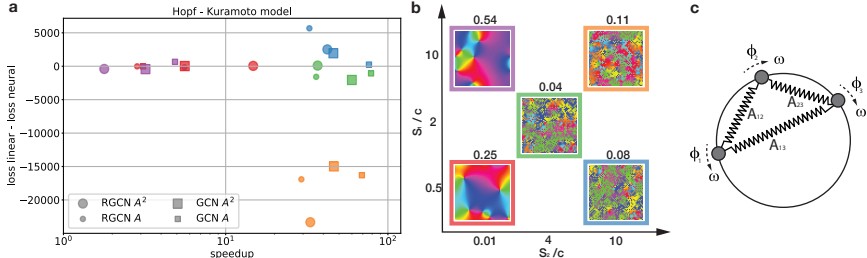

Figure 3: Hopf-Kuramoto model on a square lattice ($50 \times 50$). a) Speedup in the final loss value difference function. Points color correspond to the regions of the phase diagram (b), also, the number above each phase pattern are the global order parameters. We fixed $c = 1$ for the simplicity. c) Coupled oscillator system.

circle graph and $2.7 \pm .3$ for tree graphs. We also experimented with two layer (GCN/RGCN-2) and three layer (GCN/RGCN-3) GCNs. As expected, the overhead of deeper GCN models slows down optimization and offsets the NR speedup gains. Figure 2 (b) visualizes the evolution of the oscillator phase $\phi_i$ on a square lattice over iterations. We can see that even though different reparametrization models reach the same loss value, the final solutions are quite different. The linear version (without NR) arrives at the final solution smoothly, while GCN models form dense clusters at the initial steps and reach an organized state before 100 steps. To quantify the level of synchronization, we measure a quantity $\rho$ known as the "global order parameter" (Sarkar & Gupte (2021)): $\rho = \frac{1}{N} \left| \sum_j e^{i\phi_j} \right|$. Figure 2 (c) shows the convergence of the global order parameter over time. We can see that one-layer GCN and RGCN gives the highest amount of acceleration, driving the system to synchronization.

**Results of Hopf-Kuramoto Model.** We report the comparison of NR models on synchronizing Hopf-Kuramoto dynamics. According to the Lauter et al. (2015) paper, we identify two different main patterns on the phase diagram Fig. 9(b): ordered (small $s_2/c$, smooth patterns) and disordered (large $s_2/c$, noisy) phases. In all experiments we use the same lattice size $50 \times 50$, with the the same stopping criteria (10 patience steps and $10^{-10}$ loss error limit) and switch between the Linear and GCN model after 100 iteration steps. Fig. 3 (a) shows the loss at convergence versus the speedup. We compare different GCN models and observe that GCN with $A^2$ as the propagation rule achieves the highest speedup. Also, we can see that we have different speedups in each region, especially in the disordered phases. Furthermore, we observed that the Linear and GCN models converge into a different minimum state in a few cases. However, the patterns remain the same. (b) shows how the level of ordering changes region by region. If the global order parameter is closer to 0, we have more of a disordered phase while the parameter is closer to 1, meaning it is a more organized pattern.

We know that running dynamics in a large lattice are computationally expensive, especially close to the phase transition points where difficult to separate the slow and fast modes.

## 3.2 PERSISTENT HOMOLOGY

Persistent homology Edelsbrunner et al. (2008) is an algebraic tool for measuring topological features of shapes and functions. It provides precise definition of qualitative features, is computable with linear algebra and robust to perturbation of input data (Otter et al., 2017), see more details in Appendix B.3. An example of persistent homology is point cloud optimization Gabrielsson et al. (2020); Carriere et al. (2021). Given a random point cloud $\boldsymbol{w}$ that lacks any observable characteristics, we aim to produce persistent homological features by optimizing the position of data points.

$$\mathcal{L}(\boldsymbol{w}) = -\sum_{p \in D} \|p - \pi_\Delta(p)\|_\infty^2 + \sum_{i=1}^{n} \|\boldsymbol{w}_i - r\|^2 \tag{20}$$

where $p \in D = \{(b_i, d_i)\}_{i \in I_k}$ denotes the homological features in the the persistence diagram $D$, consisting of all the pairs of birth $b_i$ and death $d_i$ filtration values of the set of k-dimensional homological features $I_k$. $\pi_\Delta$ is the projection onto the diagonal $\Delta$ and $\sum_i d(\phi_i, S)$ constrains the points within the a square centered at the origin with length $2r$ (denote $r$ as range of the point cloud). Carriere et al. (2021) optimizes the point cloud positions directly with gradient-based optimization.

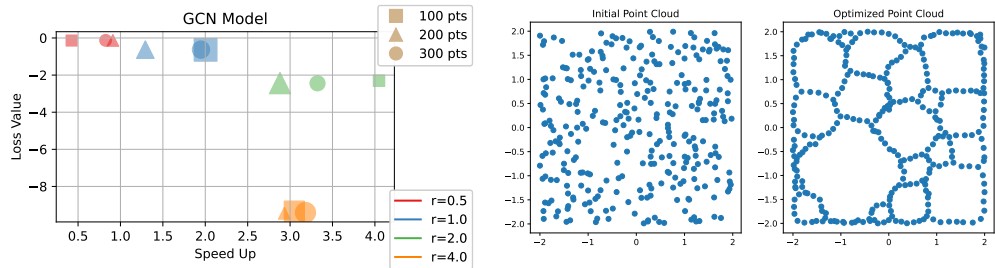

Figure 4: Speedup: a,b) Training and total time speedup; c) GCN speeds up convergence.

Figure 5: a) Loss vs speed-up of GCN model. The converged loss value is determined by point cloud range only, while the speed-up is affected by both point cloud range and size. b) Initial random point cloud and GCN-optimized point cloud.

We will refer theirs as the baseline linear model. In contrast, we integrate GCN and reparameterize the point cloud positions, and will refer ours as the GCN model.

**Implementation.** To measure the speed-up of GCN model, we used the same Gudhi library for computing persistence diagram as Gabrielsson et al. (2020); Carriere et al. (2021). The runtime of learning persistent homology is dominated by computing persistence diagram in every iteration, which has the time complexity of $O(n^3)$. Thus, the runtime per iteration for GCN model and linear model are very similar, and we demonstrate that the GCN model can reduce convergence time by a factor of $\sim 4$. We ran the experiments for point cloud of $100, 200, 300$ points, with ranges of $0.5, 1.0, 2.0, 4.0$. The hyperparameters of the GCN model are kept constant, including network dimensions. The result for each setting is averaged from 5 consecutive runs.

**Results.** Fig. 4 and 5(a) show that the speedup of the GCN model is related to point cloud density. Training converges faster as the point cloud becomes more sparse, but the speedup gain saturates as point cloud density decreases. On the other hand, time required for initial point cloud fitting increases significantly with the range of point cloud. Consequently, the overall speedup peaks when the range of point cloud is around 4 times larger than what is used be in Gabrielsson et al. (2020); Carriere et al. (2021), which spans over an area 16 times larger. Further increase in point cloud range causes the speedup to drop as the extra time of initial point cloud fitting outweighs the reduced training time. The loss curve plot in Fig. 4 shows the convergence of training loss of the GCN model and the baseline model in one of the settings when GCN is performing well. Fig. 5 shows the initial random point cloud and the output from the GCN model. In the Appendix B.3, we included the results of GCN model hyperparameter search and a runtime comparison of the GCN model under all experiment settings.

**Conclusion** We propose a neural reparametrization scheme to accelerate a large class of optimization problems. Our method is grounded on analysis that the dynamics of gradient flow are related to the condition number of the system. By reparametrizing the optimization problem with a graph convolutional network, we can modify the condition number and obtain the maximum speed up. The aggregation function of the GCN is constructed from the gradients of the loss function and reduces to the Hessian in early stages of the optimization. We demonstrate our method on optimizing synchronization problems and persistent homology of pointclouds. Depending on the experiment, we obtain a best case speedup that ranges from 4 to 80.

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
