# OpenReview forum: "Accelerating Optimization using Neural Reparametrization"
_ICLR.cc/2022/Conference — ICLR 2022 Submitted_

### Official Review · Reviewer_e51J · 2021-10-31

**Correctness:** 3
**Technical Novelty And Significance:** 2
**Empirical Novelty And Significance:** 2
**Recommendation:** 5
**Confidence:** 3

**Main Review:**

The paper introduced a new network reparametrizing method for accelerating optimization for nonlinear problems. Overall, the reviewer finds the paper is a bit hard to follow, and the presentation of the paper can be significantly improved. The experiments are interesting but the comparison is not quite comprehensive.
First, the reviewer is not fully convinced by the benefits of reparametrizaiotn. The reparameterization using a neural network can improve convergence speed, but on the other hand, the memory cost could be higher.
Second, it is a bit unclear to the reviewer why in Section 2.2, the authors considered the NTK. The weights for NTK require an ultrawide network and the weights barely change. It is a bit abrupt without much explanation of the motivations behind it.
Third, the speed up in Figure 2 does not seem impressive. The authors only compared with a very baseline optimizer. More comprehensive comparisons are needed to draw the conclusion.


**Summary Of The Paper:**

This work proposed a neural reparametrization scheme to accelerate a large class of nonconvex nonlinear optimization problems. The proposed method is grounded on analysis that the dynamics of gradient ﬂow are related to the condition number of the system. More specifically, by reparametrizing the optimization problem with a graph convolutional network (GNN), the proposed method can modify the condition number and obtain convergence speed up, the acceleration is demonstrated on optimizing synchronization problems and persistent homology of point-clouds.

**Summary Of The Review:**

Overall, the reviewer finds the paper is a bit hard to follow, and the presentation of the paper can be significantly improved. The experiments are interesting but the comparison is not quite comprehensive.

---

> ### Author Response · Authors · 2021-11-23
> **Thank you and clarifications**
>
> > Overall, the reviewer finds the paper is a bit hard to follow, and the presentation of the paper can be significantly improved.
>
> Thank you. We will work on this.
>
> > The experiments are interesting but the comparison is not quite comprehensive. First, the reviewer is not fully convinced by the benefits of reparametrizaiotn. The reparameterization using a neural network can improve convergence speed, but on the other hand, the memory cost could be higher.
>
> That is true, but we specifically tried to find a reparametrization where the memory footprint is not too large. However you are correct that, for instance, when $M$ is dense, the memory cost is significant.
>
> > Second, it is a bit unclear to the reviewer why in Section 2.2, the authors considered the NTK. The weights for NTK require an ultrawide network and the weights barely change. It is a bit abrupt without much explanation of the motivations behind it.
>
> We agree that it was not necessary to point out that $K$ is the NTK. The results about weights of NTK are derived for the case where the neural net has a large dataset as input. In our case, there is no input to the reparametrization layer, meaning those NTK results do not apply and we do not make use of them.
>
> > Third, the speed up in Figure 2 does not seem impressive. The authors only compared with a very baseline optimizer. More comprehensive comparisons are needed to draw the conclusion.
>
> We agree that more extensive experiments are needed. We have done some more hyperparameter sweeps and will work on extending the experiments.

---

### Official Review · Reviewer_WWWw · 2021-11-01

**Correctness:** 4
**Technical Novelty And Significance:** 3
**Empirical Novelty And Significance:** 3
**Recommendation:** 6
**Confidence:** 4

**Main Review:**

**Main comments**

-Overall, I think the paper is quite novel and the experiments fairly convincing.

-I really enjoy how much the authors walk through the individual steps of the gradient math which derives their neural reparamaterization in 2.1 and 2.2. It is easy to follow and clear.

-However, one drawback of this approach is that it appears that it seems to only help the early stages of optimization, as this is how it is used in the experiments. I think the authors should take more care to make this point more clear. In particular, what prevents one from using this hessian approximation for $\bar{M}$ as in Section 2.3 in early stages of training when using Adam? It would be nice to see ana ablation of the different components of their method, to understand exactly what component of the approach is contributing to the improved performance.

-How does this approach compare to gradient-based optimization in terms of memory consumption? How would this scale to large-scale datasets with larger parameter spaces, e.g. deep network training?

**Minor Points**

-The authors seem to pose the title and introduction to refer to any non-convex optimization problem, but in some parts of the paper they seem only focused on neural network optimization (e.g. Fig 1). It would be good to smooth out these inconsistencies.

-The abstract on OpenReview and the abstract in the article do not match.

-In the experiments, why is the term "linear" used to refer to the gradient-based baselines? I am not sure this is the best term to use and was confusing to me upon my first read.

**Summary Of The Paper:**

The authors derive a neural reparameterization of non-convex optimization problems in order to accelerate their convergence. They do this by deriving how the slowest components of the optimization variables can have their convergence rate improved by preconditioning with a NTK-based matrix. They make connections between this approach and Group Convolutional Networks. Experimentally, they show this approach improves upon baseline gradient-based optimization on a two datasets.

**Summary Of The Review:**

Overall, I lean slightly towards acceptance. This is due to the clarity and novelty of the paper, as well as encouraging experimental results. However, I think some more experimental verification is needed for ablating the different components of the proposed approach and for demonstrating its applicability to a broader range of problems.

---

> ### Author Response · Authors · 2021-11-23
> **Thank you and clarifications**
>
> > -I really enjoy how much the authors walk through the individual steps of the gradient math which derives their neural reparamaterization in 2.1 and 2.2. It is easy to follow and clear.
>
> We are glad if the pedagogical exposition was helpful.
>
> > -However, one drawback of this approach is that it appears that it seems to only help the early stages of optimization, as this is how it is used in the experiments. I think the authors should take more care to make this point more clear.
>
> Note that the reparametrization can be used at any stage of the optimization process. The early stage approximation with the Hessian was just a particular case in which we were able to explicitly predict what should go into the GCN, without having to calculate gradients at all. In later stages, $\kappa$ can be updated using actual gradients.
>
>
> >  In particular, what prevents one from using this hessian approximation for $\bar{M}$  as in Section 2.3 in early stages of training when using Adam? It would be nice to see ana ablation of the different components of their method, to understand exactly what component of the approach is contributing to the improved performance.
>
> The reason this is generally not done is that the matrix $\bar{M}$ can be quite dense and costly to use in $K$ at every step of optimization. The cases where our approach may work best is when $M$ is expected to be sparse. For example, when the optimization is over a sparse graph (like the experiments we used) $M$ can be sparse and $K\sim M^{-1/2}$ can also be approximated with sparse matrices. This makes it tractable for these problems, but it may not be tractable in general.
>
> > -How does this approach compare to gradient-based optimization in terms of memory consumption? How would this scale to large-scale datasets with larger parameter spaces, e.g. deep network training?
>
> We are working on understanding whether it can be used for deep networks. In terms of dataset, it won't be a problem. We have added an experiment with MNIST and can explain the details further. Essentially, the dataset only enters in the loss function $\mathcal{L}(w)$, but it does not enter the reparameterization directly. For logistic regression on MNIST (i.e. single layer neural network), the covariance matrix of the input images determines $M$ and thus $K$. The covariance can be precomputed to build the Hessian and fixing the GCN and reparametrization at the beginnign of the training.
>
> Minor Points
>
> > -The authors seem to pose the title and introduction to refer to any non-convex optimization problem, but in some parts of the paper they seem only focused on neural network optimization (e.g. Fig 1). It would be good to smooth out these inconsistencies.
>
> Note that the neural network is only required in the reparametrization. The original optimization problem does not need to be a neural network, as the Kuramoto example shows.
>
> > -In the experiments, why is the term "linear" used to refer to the gradient-based baselines? I am not sure this is the best term to use and was confusing to me upon my first read.
>
> Good suggestion. We will consider renaming it.

---

> > ### Comment · Reviewer_WWWw · 2021-11-27
> > **Thank you for the response**
> >
> > Thank you for clarifying my questions; I find these answers satisfactory, so I will keep my score as-is.

---

### Official Review · Reviewer_BGYL · 2021-11-01

**Correctness:** 3
**Technical Novelty And Significance:** 2
**Empirical Novelty And Significance:** 1
**Recommendation:** 1
**Confidence:** 3

**Main Review:**

Strengths:
* The idea of reparameterization is nice.

Weaknesses:

* The experimental evaluation consists of two problems that are not of interest to the ICLR community. I have certainly never seen either of them used in a ML paper. I have no idea how they relate to actual optimization problems I care about (i.e., training deep neural networks).
* The experimental work doesn't look thorough -- where are the learning rate sweeps, comparisons to other optimizers, etc etc?
* The paper spends a substantial of space (pg 2-4) deriving well known results (under assumptions that amount to strong convexity lambda_max to lambda_min ratio controls covergence). I strongly suggest that the authors use the results and language of optimization, rather than going from first principles for no good reason.
* The final reparameterization is not very interesting -- although much ado is made about "using a neural network parameterization", it's just a linear map at the end of the day.
* Since the reparameterization is linear, this makes the overall idea very similar to a preconditioner. This should be touched on, and compared to e.g., KFAC, shampoo, the many other linear preconditioners that people use. As with the optimization comment above, I think this work needs to be grounded more in the literature.
* GCNs are tangentially relevant, but don't seem to be used in any really meaningful way.

Technical comment: right after eqn 15, it says that H is positive semidefinite. Where does thos come from? Isn't the base problem meant to be non-convex, in which case by definition H should have some negative eigenvalues at some point?

**Summary Of The Paper:**

This paper proposes a reparameterization of non-linear non-convex optimization problems. This reparameterization amounts to a linear map (i.e., "optimization params = linear operation of a different set of parameters). These linear maps are interpreted as a graph convolution network. The experimental results are validated on "Kuramoto models" and "persistent homology models".

**Summary Of The Review:**

This paper is clearly unready for publication. The main idea -- using a structured linear reparameterization -- is under-developed, and the experimental results are on problems that the ICLR audience don't really care about.

---

> ### Author Response · Authors · 2021-11-19
> **Relation to Literature**
>
> Thank you for your helpful comments. We are trying to sharpn the narrative to contrast the actual contributions of our work in light of existing literature. Key rake-aways pertinent to your comments are:
> 1. linearity of the reparametrization is not required;
> 2. Shampoo, K-FAC and AdaGrad all work with original parameters (not reparameterizing);
> 3. in optimization on sparse graphs, GCN are especially helpful and efficient.
>
>
> > The experimental evaluation consists of two problems that are not of interest to the ICLR community. I have certainly never seen either of them used in a ML paper. I have no idea how they relate to actual optimization problems I care about (i.e., training deep neural networks).
>
> We have now also used MNIST to conduct simple experiments for logistic and linear regression (i.e. single layer networks). But we believe for deep nets methods like Shampoo may be more efficient (see below).
> Regarding the problems not being of interest, respectfully, we disagree. Our main references are both PMLR (Gabrielsson 2020 and Carriesse 2021) for persistent homology. While our method could be adapted for computer vision, etc, our main focus is scientific ML, specifically, nonlinear PDE. There are numerous examples where new use cases of ML have been introduced in AI venues, including ref Hoyer 2019 (NeurIPS workshop), DeepMind's weather now-casting, etc. We argue that Kuramoto is an attractive benchmarking problem due to its high nonlinearity, simplicity, and its importance in science and synchronization.
>
> > The experimental work doesn't look thorough -- where are the learning rate sweeps, comparisons to other optimizers, etc etc?
>
> We are adding results with different learning rates, optimizers and hidden dimensions for the $\theta$.
>
> > The paper spends a substantial of space (pg 2-4) deriving well known results (under assumptions that amount to strong convexity lambda_max to lambda_min ratio controls convergence). I strongly suggest that the authors use the results and language of optimization, rather than going from first principles for no good reason.
>
> Thank you, we will do so. We are shortening the initial derivations and refer to classic literature. One reason we needed first principles was to highlight pedagogically what changes when we reparametrize.
>
> > The final reparameterization is not very interesting -- although much ado is made about "using a neural network parameterization", it's just a linear map at the end of the day.
>
> We are now discussing more general cases, which are nonlinear neural reparamterizations.
> In the simplest case where $\mathbb{E}{M}$ and $\mathbb{E}{K}$ commute, we find the linear reparametrization, which is is closely related to the preconditioning used in AdaGrad, as we explain below.
> But, note that Theorem 1, followed from assuming independence of $K$ from $w$ (eq. 10) $\mathbb{E}[MK] = \mathbb{E}[M]\mathbb{E}[K] $.
> This is only a sufficient, but not necessary condition for optimal isotropic convergence of $\|\psi_i^T dw/dt\|$.
> It is conceivable that nonlinear reparametrizations can be found which approach optimal convergence rates.
>
>
> > Since the reparameterization is linear, this makes the overall idea very similar to a preconditioner. This should be touched on, and compared to e.g., KFAC, shampoo, the many other linear preconditioners that people use. As with the optimization comment above, I think this work needs to be grounded more in the literature.
>
> The linear reparametrization is a special case, arising from requiring $\mathbb{E}[MK]\ne \mathbb{E}[K]\mathbb{E}[K]$, which yields a preconditioner.
> In our experiments, we use nonlinear multilayer GCN instead of the linear solution.
> We will clarify in the paper that linearity is not necessary, but sufficient.
> The linear case is related to AdaGrad's full-matrix results (i.e. not just diagonal $gg^T$) as follows.
> To use AdaGrad's notation, let $g_t = \partial \mathcal{L}/\partial w(t)$ be the gradient, so that $M= gg^T $.
> In the linear case $w=\kappa \theta$ Eq. 14, we choose the optimal $\kappa= \sqrt{M/m_{max}} =$, for symmetric $\kappa$.
> Although we optimize over $\theta$, this case is equivalent to a linear preconditioning of $w$ because
> $\newcommand{\ro}{\partial}
> \newcommand{\L}{\mathcal{L}}
> \newcommand{\R}{\mathbb{R}}
> \newcommand{\eps}{\varepsilon}
> $
> \begin{align}
>     {d\theta \over dt}& = -\eps {\ro \L \over \ro w} {\ro w\over \ro \theta} =-\eps \kappa {\ro \L \over \ro w}  \cr
>     {dw \over dt}& = \kappa {d\theta \over dt}= -\eps \kappa^2 {\ro w\over \ro \theta}\cr
>     &={-\eps\over m_{max}} gg^T {\ro w\over \ro \theta}
> \end{align}
>
> So, you are correct that in our method linear reparametrization yields the classic full-matrix AdaGrad (i.e. not using diagonal $g_t^2$) as well as quasi-Newton's methods as special cases (in early stages where $gg^T \approx \mathcal{H}^2$, the above equation becomes Newton's method).

---

> ### Author Response · Authors · 2021-11-19
> **Part 2**
>
> $\newcommand{\ro}{\partial}
> \newcommand{\L}{\mathcal{L}}
> \newcommand{\eps}{\varepsilon}$
>
> However, even in the linear case, the reparametrization from $w$ to $\theta$ has benefits, as also shown in Tarmoun 2021.
> We will edit the paper to emphasize this:
> In our experiments, $w\in \R^{n\times d} $ and $\theta\in \R^{n\times h}$ are matrices and $h\gg d$.
> So $w(\theta)$ can be overparametrized, similar to Tarmoun 2021.
> Additionally, in our experiments $w(\theta)$ is nonlinear.
> While we don't have theoretical proofs for when the nonlinear case is optimal, it does perform better than linear in most cases.
>
> Regarding relation to Shampoo: Shampoo is a way to approximate the full-matrix AdaGrad with a left and right decomposition of the gradient matrices for weights of each layer.
> Hence, the results of Shampoo are a better approximation than the diagonal form of Adagrad (i.e. using $\mathrm{diag}(g_t^2)$ instead of $G = g_tg_t^T$).
> In particular, Shampoo, KFAC and all other methods still work with the original optimization variables $w$, not with any reparametrization.
> In contrast, our method is about using new variables $\theta$ and we show that under certain conditions we can take advantage of close to maximum theoretical speedup.
>
> > GCNs are tangentially relevant, but don't seem to be used in any really meaningful way.
>
> The role of GCN is actually important and two fold: 1) flexible architectures incorporating $M$; 2) Efficient for optimization problems over graphs. Let us elaborate a bit.
>
> __GCN for architecture:__
> Our task is to incorporate the square gradients $M=gg^T$ into a neural reparametrization, while still allowing free parameters $\theta$. Since $M$ is symmetric, GCN provide a natural, flexible way to construct functions of $M$ which approximate $K\sim M^{-1/2}$ needed for optimal convergence rates.
>
> __Optimization on graphs:__
> The experiments we used were on graphs. As Adagrad, Shampoo and K-FAC all point out, using the full matrix $M = gg^T$ is not computationally efficient in many cases. However, we make the point that on sparse graphs, in early stages, full matrix $gg^T$ _is in fact efficient_ to compute. The reason is, in early iterations $gg^T$ is the Hessian squared and on many graph optimization problems (specifically Reaction-diffusion) the Hessian is the graph Laplacian. For sparse graphs, multiplying a vector by graph Laplacian is $ O(n)$, hence very efficient.
> Thus, on sparse graphs GCN allows us to build functions of $M$ which are efficient enough to not slow down the iterations.
>
> >  Technical comment: right after eqn 15, it says that H is positive semidefinite. Where does thos come from? Isn't the base problem meant to be non-convex, in which case by definition H should have some negative eigenvalues at some point?
>
> There is a small error in the definition of $\mathbf{H} = \sqrt{\mathcal{H}^2/m_{max}}$. Since we are taking the square root, we need to find positive semi-definite matrices which square to $\mathcal{H}^2$. In other words, the eigenvalues of $mathbf{H}$ are aboslute values of eigenvalues of the Hessian $\mathcal{H}$, meaning $\mathbf{H}$ is PSD.

---

### Official Review · Reviewer_ue9h · 2021-11-02

**Correctness:** 3
**Technical Novelty And Significance:** 3
**Empirical Novelty And Significance:** 2
**Recommendation:** 3
**Confidence:** 3

**Main Review:**

In terms of its strength, this paper contains interesting thoughts about the intriguing idea that a temporally varying linear reparametrization of the unknown can accelerate the gradient flow based optimization. The general topic, the combination of theoretical analysis and numerical experiments, and the bridge between the two by using efficient numerical approximations of what the theory demands, are strength of this paper.  And although the numerical experiments are certainly not exhaustive, there is some proof of concept of the benefit in the particular applications considered here.

Unfortunately, the paper also has some clear drawbacks. In particular, I found the paper difficult to follow and the main idea from an optimization perspective appears to be unnecessarily hidden in a framework on "neural" reparametrizations. Unless I misunderstood the main idea significantly, the "neural reparametrization" illustrated by a neural network in Fig. 1(b), later turns out to be a linear parametrization only, i.e., considering the gradient flow for theta in $w(t) = \kappa(t) \theta(t)$ instead of in the original variable w.  Before considering this to be a graph neural network, I would have been interested in how this idea relates to other classical optimization methods: Has the idea of temporally changing but linear reparametrization not been considered in the optimization literature before? As kappa turns out to be the square root of the inverse of the Hessian, is there a relation to Newton or quasi-Newton methods? For me, the paper would have been easier to follow from this more classical optimization perspective. In particular, the considered gradient flow resulting from the linear reparametrization seems to be $\partial_t \theta(t) = \kappa(t)^T \nabla L (\kappa(t) \theta(t))$, and should be stated explicitly. If now the change in $\kappa$ is negligible slow in comparison to the change in $\theta$, and if $\kappa$ represents the (scaled) square root of the inverse Hessian, isn't that the flow arising from Newton's method? I would much rather prefer a clear motivation and presentation of the paper from such a classical perspective before delving into graph neural networks.


Some minor aspects
- In equation (2) there is an $\epsilon_{i,j}$, but I think the way eq. (1) is written it is unclear what 'off-diagonal' elements in $\epsilon$ mean. (Of course the delta ensures there are no off-diagonals, but then I would avoid the notation).
- "Equation 1 is also an ordinary differential equation"; I would call it a partial differential equation.
- I am sometimes not sure which quantities are random variables and which ones are not. In eq. (4), for instance, random variables seem to have been dragged out of the expectation, which I do not understand.
- An example of why the paper was a little difficult for me to follow are sentences like "When running GD, the maximum change in w is bounded to ensure numerical stability." This sounds like a modification of GD (like gradient clipping), but it is actually meant as a condition to limit the step size you are using. Thus, isn't the reasoning flipped, i.e., in order to ensure numerical stability, we have to bound ...?
- In the entire analysis, it could be made clearer that M is time dependent. The first sentence of section 2.3 is the first time where it is really prominent. The discretization for time dependent matrices  might of course make the behavior of the actual algorithm differ from the (continuous) gradient flow.
- Before eq. (10) it is exemplified that $w = \sigma(A\theta + b)$ would be a valid choice. $A$, $b$, and $\sigma$ are, however, not defined and if $\sigma$ refers to a (nonlinear) activation function, I do not see how this is true.
- page 2, "abounded"
- page 6, "adaptove learning rates"
- If the numerical experiments are carried out with Adam, shouldn't the theory also consider effects like (adaptively scaled) momentum?
- In Fig. 2, why does GCN-1 seemingly start with a much lower loss function value than the other methods? Does it have a sharp drop at the beginning?
- "GCN with $A^2$ as the propagation rule achieves the highest speedup". What is $A^2$? Please define.
- "where difficult to separate the slow and fast modes" >> "where it is difficult ..."
- The numerical results are, to my mind, not a strong indication of the proposed approach being a universal way to accelerate gradient-based methods. In particular, I am wondering how specific the acceleration results are to the applications? Also, what amount of hyperparameter tuning is required for the proposed approach to work well?


**Summary Of The Paper:**

This work studies the question to what extend a reparametrization of an optimization problem, i.e. representing the original parameters w to optimize for as a function of some other parameters theta, can accelerate the convergences of the gradient flow / gradient descent for nonconvex optimization problems. It studies the dynamics of the flow via eigenvectors of a matrix M formed as the expectation over the outer product of the gradient of the loss with itself to reveal 'slow' and 'fast' modes of the evolution. It subsequently derives sufficient conditions for the reparametrization (which is chosen to be linear but time varying) to balance the decay on all modes. After discussing an efficient approximation of the theoretically derived scheme, numerical results demonstrate the effectiveness of the proposed reparametrization in two exemplary applications.

**Summary Of The Review:**

Although I like the general idea and do believe that reparametrization can balance out different convergence speeds of different modes to some extend, I found the presentation to be a little confusing. The appoach seems to reduce to a linear reparametrization, which seems to relate it to other (more classical) approaches. Along with the list of minor aspects that make the paper a little difficult to follow, I need some clarification on this aspect.

---

> ### Author Response · Authors · 2021-11-19
> **Part 1: Relation to Newton and existing work**
>
> > In terms of its strength, this paper contains interesting thoughts about the intriguing idea that a temporally varying linear reparametrization of the unknown can accelerate the gradient flow based optimization. The general topic, the combination of theoretical analysis and numerical experiments, and the bridge between the two by using efficient numerical approximations of what the theory demands, are strength of this paper.
>
> Thank you for appreciating the value of our work. We are reshaping the narrative to polish it and contrast our work better with classic and existing literature on this topic.
>
> > and the main idea from an optimization perspective appears to be unnecessarily hidden in a framework on "neural" reparametrizations. Unless I misunderstood the main idea significantly, the "neural reparametrization" illustrated by a neural network in Fig. 1(b), later turns out to be a linear parametrization only, i.e., considering the gradient flow for theta in  instead of in the original variable w.
>
> We are now clarifying the point that the linear reparametrization is just one way to achieve the isotropic convergence rates we want. In practice, in our experiments we used nonlinear, multilayer GCN and the nonlinearity did provide some additional speedup. We just don't have a clear proof for what $w(\theta)$ should be in the nonlinear case and assume it is approximately similar in structure to the linear case. That is why we use GCN in the nonlinear case as well, instead of a just replacing $w$ with MLP.
>
> > Before considering this to be a graph neural network, I would have been interested in how this idea relates to other classical optimization methods: Has the idea of temporally changing but linear reparametrization not been considered in the optimization literature before?
>
> Not as far as we know. Temporally varying preconditioning is, of course, the subject of all adaptive optimizers, including AdaGrad which is closely related to our results.
> Our reparametrization is semi-supervised as it uses the square gradients $M$ and that is why we need GCN. Unsupervised reparametrization is discussed in Tarmoun 2021, among others.
>
> > As kappa turns out to be the square root of the inverse of the Hessian, is there a relation to Newton or quasi-Newton methods? For me, the paper would have been easier to follow from this more classical optimization perspective. In particular, the considered gradient flow resulting from the linear reparametrization seems to be , and should be stated explicitly. If now the change in  is negligible slow in comparison to the change in , and if  represents the (scaled) square root of the inverse Hessian, isn't that the flow arising from Newton's method?
>
> Correct, this __special case__ is almost equivalent to Newton's method, with the addition of the reparametrization to $\theta$.
> Reparametrizing can alter convergence rates (Tarmnoun 2021), potentially yielding speedup.
> If we write $dw/dt$ instead of $d\theta/dt$ we recover Newton's method (but with absolute values of the Hessian)
> $\newcommand{\ro}{\partial}
> \newcommand{\L}{\mathcal{L}}
> \newcommand{\eps}{\varepsilon}$
> \begin{align}
>     {d\theta \over dt}& =-\eps \kappa {\ro \L \over \ro w}={-\eps\over m_{max}} \mathcal{H}^{-1/2} {\ro \L \over \ro w}  \cr
>     {dw \over dt}& = \kappa {d\theta \over dt}= {-\eps\over m_{max}}  |\mathcal{H}|^{-1} {\ro w\over \ro \theta}
> \end{align}
>
> But let us emphasize that this is just a __special case__:
> In general, there may exist __nonlinear__ $w(\theta)$ which make all modes $\psi^T dw/dt$ evolve at the same rate.
> The linear reparametrization is a special case, arising from the __simplifying assumption__ that $\mathbb{E}[MK]= \mathbb{E}[K]\mathbb{E}[K]$ (eq. 10).
> The consequence of this simplification is __Theorem 1__, which is closely related to __full-matrix version of AdaGrad__, if we dynamically update gradients as $\bar{M} = \sum_t g_tg_t^T $, with $g_t= \ro \L/\ro w(\theta(t))$.
> The Hessian arises from a further simplification, assuming randomness of weights in early iterations.
> Thus our final result is not just using the Hessian. The Hessian is only used as an easy approximation for implementing our results. In the future, we will work on dynamically updated reparametrization.
>
> >  I would much rather prefer a clear motivation and presentation of the paper from such a classical perspective before delving into graph neural networks.
>
> One Key point which we will try to clarify is that the Hessian result is especially relevant for optimization problems on sparse graphs.
> In this case, the Hessian usually becomes the graph Laplacian. While computing its inverse to use Newton's method and multiplying it in every step is quite expensive, GCN allows us to approximate the process very efficiently ($O(kn)$) in sparse graphs.
> Additionally, n our experiments we used __nonlinear__ GCN models, which yielded some additional speedup.

---

> ### Author Response · Authors · 2021-11-23
> **Part 2**
>
> $\newcommand{\ro}{\partial}
> \newcommand{\L}{\mathcal{L}}
> \newcommand{\eps}{\varepsilon}$
>
> > Some minor aspects
> > In equation (2) there is an , but I think the way eq. (1) is written it is unclear what 'off-diagonal' elements in  mean. (Of course the delta ensures there are no off-diagonals, but then I would avoid the notation).
>
> Note that, in the original AdaGrad results the learning rates are given by a matrix $M=\sum_t g_tg_t^T$, where $g_t=\ro\L/\ro w(t)$. This matrix is generally considered intractable to compute for every batch and so the diagonal version $\sum_t \mathrm{diag}(g_t^2)$ is used instead. Adam and RMSProp generalize this slightly and add momentum, but still use the diagonal version. Shampoo, for instance, does have off-diagonal terms, though not as much as the full AdaGrad. Thus, in the full matrix version of AdaGrad $\eps_{ij}= \eta [M^{-1/2}]_{ij}$ does have off-diagonal terms.
>
> > "Equation 1 is also an ordinary differential equation"; I would call it a partial differential equation.
>
> Correct, we will fix that.
>
> > I am sometimes not sure which quantities are random variables and which ones are not. In eq. (4), for instance, random variables seem to have been dragged out of the expectation, which I do not understand.
>
> We apologize, the derivatives need to corrected. We are Taylor-expanding $\L(w)$ around a given point $w(0)$, which means the partial derivatives are w.r.t. $w(0)$, not $w$ .
>
> $$ \L(w(0)+w) = \L(w(0))+ w\cdot \left.{\ro \L(v)\over \ro v}\right|\_{v\to w(0)} + \left.{1\over 2}\sum_{ij} w_i w_j  {\ro^2 \L(v)\over \ro v_i\ro v_j}\right|_{v\to w(0)} + \dots
> $$
> Then, $w$ are instances of a random variable, while $w(0)$ is fixed. Therefore, in eq. 4, the partial derivatives can come out of expectations.
>
>
> > An example of why the paper was a little difficult for me to follow are sentences like "When running GD, the maximum change in w is bounded to ensure numerical stability." This sounds like a modification of GD (like gradient clipping), but it is actually meant as a condition to limit the step size you are using. Thus, isn't the reasoning flipped, i.e., in order to ensure numerical stability, we have to bound ...?
>
> Correct, we meant the latter: for numerical stability, we limit the step size to ensure the total change in $w$ is bounded. We will change the wording to remove confusion.
>
> > In the entire analysis, it could be made clearer that M is time dependent. The first sentence of section 2.3 is the first time where it is really prominent.
>
> Great point, we will do that.
>
> > Before eq. (10) it is exemplified that $w=\sigma(A\theta + b)$ would be a valid choice. $A$, $b$, and $\sigma$ are, however, not defined and if $\sigma$ refers to a (nonlinear) activation function, I do not see how this is true.
>
> We will clarify this. We meant for arbitrary $A,b$ and nonlinear $\sigma$. For the general nonlinear $w(\theta)$, the expected mode convergence rate is $\|\psi_i^Tdw/dt\|$ is difficult to compute because $\mathbb{E}[KMK^T]$ cannot be separated into $\bar{K}$ and $\bar{M}$.
> For example, for 1D $w$, if $w = \theta^2$, and $\L=w^2$, we have $K= 4w$ and $M=4w^2=K^2/4$, meaning they are not independent.
> This _does not_ mean that no nonlinear $w(\theta)$ would yield optimal speedup.
> The linear case, on the other hand, simplifies because $\mathbb{E}[MK]= \mathbb{E}[K]\mathbb{E}[K]$ and can be solved explicitly. The resulting Theorem 1 is related to AdaGrad, because it states
>
> \begin{align}
>     %\kappa & = \sqrt{{M\over m_{max}}} = \sqrt{{gg^T\over m_{max}}} \cr
>     %g &\equiv {\ro \L \over \ro w} \approx \mathcal{H}^{-1/2}\cr
>     {d\theta \over dt}& = -\eps {\ro \L \over \ro w} {\ro w\over \ro \theta} =-\eps \kappa {\ro \L \over \ro w}  \cr
>     {dw \over dt}& = \kappa {d\theta \over dt}= -\eps \kappa^2 {\ro w\over \ro \theta}\cr
>     &={-\eps\over m_{max}} gg^T {\ro w\over \ro \theta}
> \end{align}
>
> Thus, the general nonlinear reparametrization reproduces AdaGrad as a special case, when $w(\theta)$ is linear. It  further reduces to Newton's method when $w(0)$ are random Gaussian.
>
> > If the numerical experiments are carried out with Adam, shouldn't the theory also consider effects like (adaptively scaled) momentum?
>
> Good Question. The effect of momentum is that the gradient used in GD is not just the instantaneous gradient, but rather some moving average over past gradients.
> This means that $M_t\approx \sum_t \gamma^{-m} g_{t-m}g_{t-m}^T$. So momentum only redefines $M$ and the rest of the analysis should not be affected.
> Especially since we do not want to update $K$ frequently, presence of momentum should not play a major role.

---

> ### Author Response · Authors · 2021-11-23
> **part 3**
>
>
> > In Fig. 2, why does GCN-1 seemingly start with a much lower loss function value than the other methods? Does it have a sharp drop at the beginning?
>
> The output of our GCN (without residual connection) is expected to have a different loss than random initialization from the beginning.
> We are adding the derivation of this to the appendix. In short, at the beginning the randomness of $w\in \R^{n\times d\_w}$ and $\theta\in \R^{n\times d_\theta}$ makes the loss approximately quadratic, expressed using only the Hessian $\mathcal{H}\_{ij} = \ro\L/\ro w_i \ro w_j$.
> Before reparametrization at $t=0$ and with each row initialized as $w_i(0)\sim \mathcal{N}(0,\sigma_w)$ we have
> $\newcommand{\H}{\mathcal{H}}$
> \begin{align}
>     \L &\approx {1\over 2} \mathrm{Tr}\left[w^T \H w \right] = {1\over 2} \sum_{i,j,k} w_{ik} \H_{ij} w_{jk} \\\\
>     & \approx {\sigma_w d_w \over 2} Tr[\H]
> \end{align}
> After reparametrization, with $\theta_a(0)\sim \mathcal{N}(0,\sigma_\theta )$
> \begin{align}
>     w & = \kappa \theta = \sqrt{h_{max}} \H^{-1/2} \theta \\\\
>     \L &\approx {1\over 2} \mathrm{Tr}\left[w^T \H w \right]  \approx {\sigma_\theta d_\theta h_{\max}\over 2}  \mathrm{Tr}[\H^{-1/2}\H \H^{-1/2}] \\\\
>     & = {\sigma_\theta d_\theta h_{\max}\over 2} \mathrm{Tr}[\H |\H|^{-1}]
>     %= {1\over 2} \sum_{i,j,k} w_{ik} \H_{ij} w_{jk} \\
> \end{align}
> depending on the initializationa and the number of negative eigenvalues of the Hessian $\H$
>
>
>
>
> > "GCN with $A^2$ as the propagation rule achieves the highest speedup". What is $A^2$? Please define.
>
> $A$ is the adjacency matrix of the graph on which we run the Kuramoto-Hopf model, defined earlier in sec. 3.1, and $A^2 = A\cdot A$ is its square. Propagation rule is the aggregation rule.
>
> > The numerical results are, to my mind, not a strong indication of the proposed approach being a universal way to accelerate gradient-based methods. In particular, I am wondering how specific the acceleration results are to the applications?
>
> That is a good point. We also believe that our method is particularly useful for optmization problems on sparse graphs. It may need further adjustments for deep neural networks.
>
> > Also, what amount of hyperparameter tuning is required for the proposed approach to work well?
>
> Generally, it didn't require much searching. You can find plots for different width of GCN layers in the appendix. We have now also added results for varying learning rates for the persistent homology experiments.
> Note that this experiment has one pretraining phase, where we fit the output of the GCN to the point-cloud to be optimized. As expected, when the learning rate for pretraining is higher, we get more speedup.
> Also, when the learning rate of the final optimization is smaller, GCN yields more speedup.

---

### Author Response · Authors · 2021-11-23
**General Comment**

We wish to thank all the reviewers for their insightful comments.
Thank you for pointing out important literature which we missed in the initial version of the paper.
Your comments also helped us reshape the narrative and sharpen the message about the actual contribution of our paper.
We also agree that it is essential to further investigate the tractability of our method in important problems such as deep neural networks.
We have done some preliminary tests with logistic and linear regression on MNIST, but the speedup is marginal. We have also done parameter sweeps and the range of the observed speedup is similar to what was reported originally (see appendix).
Finally, note that while our theoretical results were on linear reparametrization, our experiments used nonlinear activations, which provided a slight performance boost.

Based on what have learned so far in this work, we believe the contributions are as follows:
1. Neural reparametrization provides a flexible method for approaching the ideal convergence rates, derived in the AdaGrad paper (the full-matrix form of it where $M = \sum_t g_tg_t^T$, with $g_t = \partial\mathcal{L}/\partial w(t)$.)
2. In the simplest case, the reparametrization becomes linear and is equivalent to full matrix Adagrad (see comments below)
3. Further simplification is poosble in early stages, reducing $M$ to the Hessian.
4. The Hessian approximation of the linear case is related to quasi-Newton's method, but more flexible due to $\theta$ and architecture of the neural reparametrization.
4. In optimization problems on sparse graphs, the Hessian preconditioning (e.g. full matrix AdaGrad) can be efficiently implemented using GCN, resulting in significant speedup.

The Key take-aways are
* Nonlinear neural reparametrization is quite general. Linear preconditioning, Adaptive gradients and quasi-Newtonian methods all arise as special cases of it.
* Neural reparametrization can be tractable in certain problems, including optimization on sparse graphs, and it can yield significant speedup.

To highlight these points, we will make specific changes for future versions:

1. We add a discussion of and emphasize the differences with Shampoo and other relevant preconditioning methods,
2. We discuss and show the relation to quasi-Newton's method
3. We discuss how the neural reparametrization is more genera than preconditioning and derive Preconditioning and Newton's method as special cases of our method.
3. For the nonlinear reparametrization, we argue that the GCN is a nonlinear preconditioner which is dynamically updated and smoothly modifies the Hessian case.
4. For optimization on graphs, we derive the conditions where using the Hessian approximation will be feasible, thus showing that full-matrix AdaGrad in early iterations can be feasible with potentially large speedups.
5. We connect more to classic optimization literature and use references to replace some of the existing results.

---

> ### Comment · Reviewer_ue9h · 2021-11-26
> **Thank you for the clarifications and the proposed changes**
>
> I'd like to thank the authors for their very detailed responses and clarifications, that are very helpful for my understanding of their work. While to my mind the current version of this paper is not quite ready to be accepted as is, I do believe that a majorly revised version that includes the 'changes for future versions' highlighted above would be a very promising, interesting, and original research paper!

---

### Decision · Program_Chairs · 2022-01-20

**Decision:**

Reject

**Comment:**

This paper proposes speeding up certain optimization problems common in physics by reparameterizing their parameters as the output of a graph neural network. The reviewers appreciate the idea, but are not convinced enough to recommend the paper for acceptance. They point out the following weaknesses:
* The method amounts to linear preconditioning, and hence it's reasonable to expect a fairly complete comparison to the many linear preconditioning approaches that have been proposed previousl. The reviewers are not satisfied with the currently provided comparison.
* The main idea is not presented clearly enough. In particular, it's not obvious the proposed method is best described as neural reparameterization, since it seems to amount to linear preconditioning.
* The experiments are not persuasive enough: The presented problems may not be relevant to all of the target audience of ICLR, and the experimental evaluation does not seem sufficiently exhaustive.

The suggested areas of improvement provided by the reviewers seem reasonable to me: I therefore recommend not accepting the paper in its current form. To make the paper more accessible and appealing, the authors may consider rewriting the paper to more closely match the perspective taken by the reviewers, and to provide a more thorough comparison to the previous approaches and the existing literature.